# CROSS-TASK GRADIENT HARMONIZATION FOR META-LEARNING

## ABSTRACT

We introduce Dynamic Gradient Harmonization, a novel solution to the gradient conflict issue in optimization-based meta-learning. Meta-learning's goal is to adapt quickly to unseen tasks with limited training examples. A number of meta-learning strategies aim to identify an optimal model initialization, subsequently updating the meta-model interactively using gradients from adapted models fine-tuned on a variety of tasks. However, existing methods neglect potential conflicts among meta gradient updates from different tasks, hindering the meta-model's training. In response to this shortcoming, we propose a dynamic gradient harmonization technique. Our proposed technique harmonizes these conflicting gradient updates, enabling a unified, effective meta-model update. This is achieved by computing a primary gradient update from weighted aggregation of gradients from fine-tuned models, using an attention operator to emphasize the primary gradients. We also implement an explore-exploit mechanism to prevent over-commitment to local optima. Experimental results demonstrate the effectiveness of our approach, resulting in more efficient training and improved generalization to new tasks.

## 1 INTRODUCTION

Meta-learning has shown to be effective in solving the problem of learning with limited training examples. Popular deep learning models severely rely on the quantity and quality of training examples, which suffer from under-fitting or over-fitting problems, especially with insufficient training data. However, in practice, large-scale data are not always available. Meta-learning tries to solve this problem by assuming that some data samples share the same set of base features though seem unrelated Li et al. (2017); Ren et al. (2018); Munkhdalai & Yu (2017). Based on this assumption, a model can be pre-trained to encode these base features and quickly transferred to learn new features with limited new data. A number of meta-learning strategies aim to identify an optimal model initialization, subsequently updating the meta-model interactively using gradients from adapted models fine-tuned on a variety of tasks. Model-Agnostic Meta-Learning (MAML) Finn et al. (2017) is the most representative method. Unfortunately, these prior optimization-based methodologies tend to aggregate gradients equally from different tasks to update the meta-model. This approach overlooks potential conflicts between gradient updates across tasks, which can obstruct the effective training of the meta-model Liu et al. (2021).

The gradient conflict problem is a well-researched topic in the multi-task learning community. Existing works solve this problem in multi-task learning by directly manipulating gradients from each well-tuned model to form a better update gradient Yu et al. (2020); Chen et al. (2020). However, unlike multi-task learning, meta-learning will focus on a general meta-objective rather than a fixed set of predefined tasks. Therefore, directly applying methods in multi-task learning is not an optimal solution to the gradient conflict problem in meta-learning with different optimization objectives.

To address the gradient conflict problem in meta-learning, we propose a metric called Gradient Conflict Index, which can reflect the divergence across different tasks. Furthermore, inspired by this metric, we introduce a novel approach, termed Gradient Harmonization, to mitigate the gradient conflict issue that emerges across tasks in meta-learning. In particular, we propose using weighted gradients aggregation to fuse gradients from various tasks. The weights of fine-tuned models are computed by comparing their gradients with the average gradients of all fine-tuned models. In this way, the gradients from fine-tuned models closer to the average gradients will contribute more to

updating the meta-model. Since the gradient conflicts can be different across layers in a neural network, we propose a layer-wise dynamic gradient harmonization method, which employs an attention layer to dynamically aggregate gradients from different fine-tuned models for each layer. To avoid over-emphasizing gradients from some specific fine-tuned models, we propose an explore-exploit learning schedule with cosine annealing, allowing the meta-model to balance exploration and exploit. The experimental results on various datasets demonstrate the effectiveness of our methods.

## 2 RELATED WORK

This section discusses related works to meta-learning and gradient conflict.

### 2.1 META-LEARNING

Meta-learning algorithm aims to train a model that can quickly adapt to unseen tasks. A meta-learning algorithm consists of an inner loop that trains a model independently on different tasks to get different well-tuned models and an outer loop that evaluates these well-tuned models on query data and aggregates all gradients to update the base model. Note that meta-learning fundamentally differs from multi-task learning, which trains neural networks on multiple tasks simultaneously. Multi-task learning only focuses on certain tasks. However, meta-learning will focus on the meta-objective presented in the outer loop. The meta-objective can be very general, such as letting the model learn a learning process on certain tasks.

Currently, the work on meta-learning can be divided into three categories:metric-based, model-based, and optimized-based methods Huisman et al. (2021). The metric-based and model-based meta-learning algorithms all introduce extra features and models Koch et al. (2015); Vinyals et al. (2016); Snell et al. (2017); Sung et al. (2018); Shyam et al. (2017); Santoro et al. (2016); Munkhdalai & Yu (2017); Mishra et al. (2017). The optimization-based meta-learning algorithms, on the other hand, modify the traditional neural network's optimization steps to make it feasible for few-shot learning. MAML Finn et al. (2017) is an important algorithm in this category. The key idea is to get a good initialization parameter set $\theta$ to quickly adapt to the unseen task. The inner loop fine-tunes the meta-model on support data. The outer loop evaluates these fine-tuned models on query data gradient to update the original base model. Meta-SGD Li et al. (2017) is similar to MAML, except it will learn a specific learning rate for each parameter in the meta-model using a multi-layer-perceptron network.

### 2.2 GRADIENT CONFLICT PROBLEM

In learning problems with multiple tasks, the base model is fine-tuned by each task with its updated gradient. The base model will be updated as below:

$$\boldsymbol{g}' = \boldsymbol{\theta} - \alpha \frac{1}{K} \sum_{T_i} \boldsymbol{g}_i, \tag{1}$$

where $k$ is the number of tasks and $\boldsymbol{g}_i$ is the gradient associated with the task $T_i$. However, when these gradients $\boldsymbol{g}_i$ do not align well, the final $\boldsymbol{g}'$ cannot reflect the optimal update direction and step.

Several works have been proposed to tackle the gradient conflict problem in the multi-task learning setting, Yu et al. (2020) directly manipulates the conflicting gradients by projecting one gradient on the normal plane of the other gradient to reduce the angle and magnitude difference between two gradients. Chen et al. (2020) propose a "GradDrop" algorithm, which constructs a mask for each gradient by measuring how many positive gradients are presented at a certain value. Liu et al. (2021) try to pick a gradient that can decrease not only the average loss but also the loss for every specific task. In the meta-learning area, Jerfel et al. (2019) try to grasp the relatedness between different tasks and cluster the similar tasks together. Yao et al. (2019) cluster the tasks into different groups and apply knowledge adaption to transfer the knowledge to cluster-based initialization. Unlike task-similarity-based methods Yao et al. (2019); Jerfel et al. (2019), this work propose to resolve the conflicts existing among these fine-tuned models. Updating the base model with these conflicting gradients will compromise the model performance. SHI et al. (2023) propose to measure the pairwise confliction severity of each layer of the model and set the layer with the largest severity as the task-dependent layer. Under meta-learning setting, these method will be less effective since they are designed to optimize a fixed set of tasks not a general objective. To address this issue, we propose a weighted gradient aggregation method, which aggregate gradients with different weights to focus on better generalized fine-tuned models.

## 3 CROSS-TASK GRADIENT HARMONIZATION FOR META-LEARNING

In this section, we propose a metric to measure the Gradient Harmonization method designed to mitigate the cross-task gradient conflict issue inherent in optimization-based meta-learning.

### 3.1 GRADIENT CONFLICT PROBLEM IN OPTIMIZATION-BASED META LEARNING

In optimization-based meta-learning, *gradient conflict* refers to the misalignment of gradients across distinct tasks during the meta-update phase Liu et al. (2021). Algorithms such as MAML (Model-Agnostic Meta-Learning) Finn et al. (2017) typically employ a two-level optimization process. This involves an inner loop that adapts to each task individually and an outer loop that updates the meta-model based on the performance of the adapted models. Specifically, in the outer loop, gradients from the adapted models across different tasks are aggregated to update the meta-model. However, when tasks are diverse and their gradients point in various directions, the cumulative update may not be optimal for any single task. This can result in suboptimal training of the meta-model, a challenge known as gradient conflict.

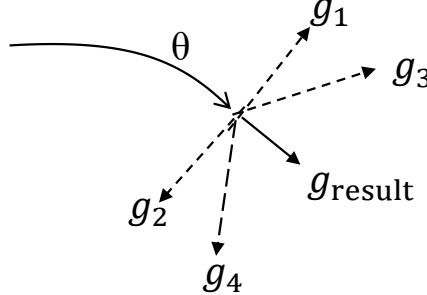

Figure 1: Illustration of gradient conflict. When updating the meta-model $\boldsymbol{\theta}$ for four tasks, conflicts emerge among the gradients derived from each task, thereby reducing the efficiency of the update. For instance, gradients $\boldsymbol{g}_1$ and $\boldsymbol{g}_2$ are in opposing directions.

Formally, in each meta-update cycle, an optimization-based meta-learning algorithm randomly selects a batch of tasks and refines a unique model for each task. Subsequently, the gradients of each fine-tuned model are aggregated to update the meta-model. The typical meta-update rule is as follows:

$$\boldsymbol{\theta}^t = \boldsymbol{\theta}^{t-1} - \beta \nabla_{\boldsymbol{\theta}^{t-1}} \sum_{T_i} \mathcal{L}_{T_i}(\boldsymbol{\theta}_i^{t-1}) = \boldsymbol{\theta}^{t-1} - \beta \sum_{T_i} \nabla_{\boldsymbol{\theta}^{t-1}} \mathcal{L}_{T_i}(\boldsymbol{\theta}_i^{t-1}). \tag{2}$$

In the equation, $\boldsymbol{\theta}^t$ represents the meta-model at the $t$th step, $\boldsymbol{\theta}_i^{t-1}$ denotes the $i$th model fine-tuned on task $T_i$, and $\mathcal{L}_{T_i}(\boldsymbol{\theta}_i^{t-1})$ is the loss incurred when $\boldsymbol{\theta}_i^{t-1}$ is evaluated on task $T_i$. $\beta$ is the learning rate employed for training the meta-model. Gradient conflict occurs when the gradients $\nabla_{\boldsymbol{\theta}^{t-1}} \mathcal{L}_{T_i}(\boldsymbol{\theta}_i^{t-1})$ point in diverse directions. An example of this gradient conflict is depicted in Figure 1. Resolving such conflicts is crucial for achieving more efficient meta-learning optimization.

The gradient conflict issue is also observed in multi-task learning, yet the two scenarios are fundamentally distinct owing to their differing learning objectives. Multi-task learning focuses on training a model on a fixed set of tasks; essentially, it employs a single-level optimization process devoid of a meta-objective. Techniques Liu et al. (2021); Yu et al. (2020); Chen et al. (2020) designed to resolve gradient conflicts in this context aim to enhance performance on these predefined tasks, which may not translate effectively to the meta-learning setting. In meta-learning, the objective is to effectively solve unseen future tasks Hospedales et al. (2020). As such, methods successful in multi-task learning may not necessarily be applicable or effective in the context of meta-learning. Consequently, it's crucial to develop and refine methods specifically designed to address gradient conflict in the realm of optimization-based meta-learning.

### 3.2 QUANTITATIVE METRIC FOR GRADIENT CONFLICT

To quantify the extent of gradient conflict in meta-learning, a suitable metric is essential. This will further inspire our proposed methods and facilitate experimental evaluations. Existing metrics SHI et al. (2023) primarily assess conflicts via pairwise gradient divergence, which is not suitable for measuring conflict among multiple gradients. To overcome this shortcoming, we introduce a new metric, the Gradient Conflict Index (GCI). This index, inspired by variance metrics, is capable of measuring divergence.

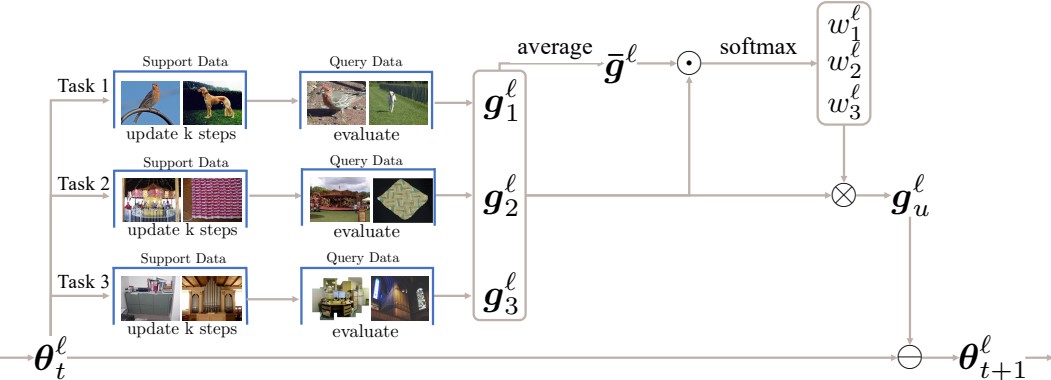

Figure 3: An illustration of the proposed layer-wise dynamic gradient harmonization method. Given gradients of three fine-tuned models, the update gradients are computed separately for each layer. For the $\ell^{th}$ layer, the query gradients are computed by taking the weighted average of gradients from the corresponding layers of three fine-tuned models. Then, an attention operator is used to dynamically fuse the update gradients.

Specifically, given gradients from a batch of tasks where $g_i$ is from task $T_i$, we first compute the aggregated gradient $\bar{g}$. Subsequently, the GCI value is calculated as follows:

$$\text{GCI}(g_i, \bar{g}) = \sum_{T_i} ||g_i||(1 - \cos(g_i, \bar{g})) = \frac{||g_i|| - g_i \cdot \bar{g}}{||\bar{g}||}, \qquad (3)$$

where $\cos(\cdot)$ is the cosine similarity function. This metric effectively captures both the divergence in magnitudes and task gradients' directions. When task gradients exhibit significant directional divergence, the GCI value increases due to higher values of $1 - \cos(g_i, \bar{g})$. Similarly, the GCI value rises when there is a greater divergence in magnitude. Figure 2 depicts the relationship between task loss and the GCI value for 100 randomly selected tasks from the Mini-ImageNet dataset, a benchmark for meta-learning-based few-shot classification. Each batch consists of four 5-way-1-shot image classification tasks. The figure demonstrates that batches with lower GCI values correspond to reduced gradient conflicts, resulting in enhanced training performance.

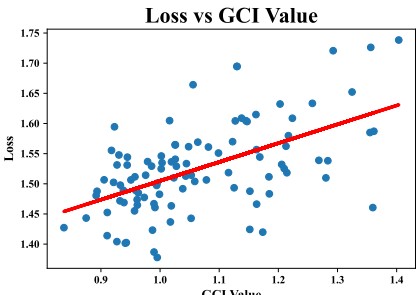

Figure 2: Loss vs. GCI values. The red line is the regression line.

### 3.3 Layer-Wise Dynamic Gradient Harmonization

Guided by the proposed GCI metric, it becomes crucial to prioritize task gradients that closely align with the aggregated update gradients, both in magnitude and direction, to mitigate gradient conflict. Consequently, we propose a Gradient Harmonization algorithm that assigns varying weights to gradients from different tasks. Specifically, it places greater emphasis on gradients from fine-tuned models that more closely resemble the final update.

We implement gradient harmonization by attributing different weights to various gradient updates to accomplish this. These weights are assigned based on each update's similarity to the average meta-update. Also, when updating the meta-model, weights across the different layers of a neural network are conventionally updated using the same weights. However, it has been observed that the degrees of conflict can vary across different layers. A uniform update rule for the entire network may not yield the optimal solution for all layers. We propose a layer-wise dynamic gradient aggregation method utilizing an attention operator to address this issue. This operator is designed to dynamically emphasize the most significant gradients for each layer. Given a set of $K$ gradients $[g_1^\ell, \ldots, g_K^\ell]$ for layer $\ell$, we first calculate the query gradient $g_q^\ell$ by averaging these gradients. We then employ an

attention operator with $\boldsymbol{g}_q$ as the query vector and $\boldsymbol{g}^\ell$ as both key and value vectors. We use the GCI values as the aggregation coefficients. This process is formally defined as follows:

$$\bar{\boldsymbol{g}}^\ell = \frac{1}{K} \sum_{k=1}^{K} \boldsymbol{g}_k^\ell, \ \ w_i' = \frac{\exp(-\text{GCI}(g_i, \bar{g}))}{\sum_{k=1}^{K} \exp(-\text{GCI}(g_k, \bar{g}))}, \ \boldsymbol{g}_u^\ell = \sum_{k=1}^{K} w_k' \boldsymbol{g}_k^\ell, \ \boldsymbol{\theta}^{\ell+1} = \boldsymbol{\theta}^\ell - \beta \boldsymbol{g}_u^\ell. \quad (4)$$

This process fundamentally employs an attention operator. This operator can adaptively adjust the aggregation weights for different layers, so it dynamically resolves varying degrees of conflict based on the GCI metric. The specific details of our proposed layer-wise dynamic gradient harmonization algorithm are presented in Algorithm 1. SHI et al. (2023) observes that the confliction score can be very different between different layers in a model. Therefore, we apply the layer-wise attention operator to adapt to different confliction score of each layer in the model. Figure 3 provides a visual example of the proposed method in action.

### 3.4 EXPLORE-EXPLOIT LEARNING SCHEDULER

Previously, we introduced a Dynamic Gradient Harmonization approach aimed at resolving gradient conflicts by emphasizing certain fine-tuned models. While this method effectively mitigates gradient conflicts, it risks over-emphasizing certain models, potentially leading to sub-optimal solutions due to insufficient exploration of the solution space. To address this concern, we propose an Explore-Exploit learning schedule (EE) to ensure a thorough exploration of the solution space. During the exploitation phase, the meta-model is updated using the proposed layer-wise dynamic gradient aggregation strategy. Conversely, during the exploration phase, the original update strategy is employed, allowing the meta-model to explore solutions by gradients of less-emphasized models.

Drawing inspiration from the cosine annealing learning rate schedule Loshchilov & Hutter (2016), we incorporate the exploration step at a probability that follows cosine annealing cycles. The probabilities of executing explore and exploit steps are calculated as follows:

---

**Algorithm 1** Dynamic Gradient Harmonization

1: **Require:** A task distribution $T$, inner loop learning rate $\alpha$, inner loop update steps $k$, outer loop learning rate $\beta$
2: Initialize model parameter $\boldsymbol{\theta}$
3: **while** not converge **do**
4:     Sample a subset of tasks $T_i$ from $T$
5:     **for** all task in $T_i$ **do**
6:         **for** inner loop step $k$ **do**
7:             Sample $D = (x_i, y_i)$ from $T_i$ as support
8:             Evaluate $L_k^D(\boldsymbol{\theta}_i)$
9:             update $\boldsymbol{\theta}_i = \boldsymbol{\theta}_i - \alpha * \nabla_{\boldsymbol{\theta}_i} L_k^D(\boldsymbol{\theta}_i)$
10:         **end for**
11:         Use $D' = T_i - D$ as query
12:         $g = \nabla_{\boldsymbol{\theta}_i} L_t^{D'}(\boldsymbol{\theta}_i)$
13:     **end for**
14:     **if** explore **then**
15:         perform the original meta-update
16:     **else**
17:         **for** all layer $\ell$ **do**
18:             **for** all task T in meta-model **do**
19:                 $\boldsymbol{g}_u^\ell = \text{Attn}\left(\frac{1}{K}\sum_{k=1}^{K}\boldsymbol{g}_i^\ell, \boldsymbol{g}^\ell, \boldsymbol{g}^\ell\right)$
20:             **end for**
21:             $\boldsymbol{\theta}^\ell = \boldsymbol{\theta}^\ell - \beta \boldsymbol{g}_u^\ell$
22:         **end for**
23:     **end if**
24: **end while**

---

$$p_{\text{explore}}^e = p_{min} + \frac{1}{2}(p_{max} - p_{min})(1 + \cos(\pi e/E)), \ \ p_{\text{exploit}}^e = 1 - p_{\text{explore}}^e,$$

where $p_{max}$ and $p_{min}$ denote the maximum and minimum probabilities of initiating the exploration step, respectively. The term $e$ signifies the current epoch number, while $E$ represents the transition epoch number, marking the point at which the trend in probabilities alters.

By employing the cosine annealing schedule, the probabilities of exploration follow a cosine function. The value of $e$ determines when the probability reaches $p_{max}$ or $p_{min}$. In accordance with this schedule, the meta-model is inclined to explore during the early stages of training to establish essential knowledge about the learning space. As training advances, the meta-model tends to exploit this knowledge for more efficient optimization. Consequently, the meta-model achieves a balance between exploration and exploitation, ensuring effective and efficient learning.

## 4 EXPERIMENTS

We conduct experimental studies on different real-world datasets to evaluate our proposed methods.

Table 1: Comparison results for 5-way classification on the Mini-ImageNet dataset. The backbone model used for gradient harmonization is Convnet4. All results of the baseline models are reported from their original works. The result of multi-task methods are reproduced from official released codes. "-" indicates results not available.

| Model | Type | 5-way-1-shot | 5-way-5-shot |
|---|---|---|---|
| Matching Network Vinyals et al. (2016) | metric | 43.44±0.77 | 55.31±0.73 |
| Relation Network Sung et al. (2018) | metric | 50.44±0.82 | 65.32±0.70 |
| Prototype Network Snell et al. (2017) | metric | 49.42±0.78 | 68.20±0.66 |
| GNN Garcia & Bruna (2017) | metric | 50.33±0.36 | 66.41±0.63 |
| MAML+PC-GRAD Yu et al. (2020) | multi-task | 50.93±1.88 | 65.93±0.95 |
| MAML+GRAD-DROP Chen et al. (2020) | multi-task | 20.37±1.45 | 23.83±0.65 |
| MAML+CA-GRAD Liu et al. (2021) | multi-task | 50.03±1.88 | 64.32±1.05 |
| Meta-Learner LSTM Ravi & Larochelle (2016) | optimization | 43.44±0.77 | 60.60±0.71 |
| TAML Jamal & Qi (2019) | optimization | 46.28±0.79 | 62.92±0.66 |
| MAML Finn et al. (2017) | optimization | 48.17±1.75 | 63.11±0.91 |
| ANIL Raghu et al. (2019) | optimization | 46.70±0.40 | 61.50±0.50 |
| BOIL Oh et al. (2020) | optimization | 49.61±0.16 | 66.45±0.37 |
| Meta-SGD Li et al. (2017) | optimization | 50.47±1.87 | 64.03±0.94 |
| HSML  Yao et al. (2019) | optimization | 50.38±1.85 | - |
| Reconciling Jerfel et al. (2019) | optimization | 49.60±1.50 | 64.60±0.92 |
| MAML+LWDGH (ours) | optimization | **51.40**±1.76 | **67.06**±0.97 |

## 4.1 EXPERIMENTAL SETTINGS

This section introduces the datasets and backbone models to evaluate our proposed methods.

**Dataset**
We evaluate the proposed methods on three datasets: Mini-ImageNet Gidaris & Komodakis (2018), Tiered-ImageNet Lee et al. (2019), and Cub200 Welinder et al. (2010) datasets. The *Mini-ImageNet* dataset consists of 100 classes with 600 samples per class. Each data sample is a $84 \times 84$ colored image. By following previous works Ravi & Larochelle (2016), we split 100 classes into 64, 16, and 20 class groups for training, validation and testing. The *Tiered-ImageNet* dataset consists of 608 classes with 779,165 images. These 608 classes are furthered combined into 34 high-level classes. These high-level classes are divided into 20, 6, and 8 class groups for training, validation and testing Ren et al. (2018). The *Cub200* dataset contains 11,788 images with 200 different labels of birds. These data are divided into 100, 50, and 50 class groups for training,validation and testing Ye et al. (2020).

**Backbone Model**
We use two backbone models in our experiments: Convnet4 and Resnet12 He et al. (2016). *Convnet4* is a 4-layer Convolutional Neural Network with 32.9k trainable parameters. *Resnet12* is a 18-layer Convolutional Neural Network with 8 million trainable parameters. Using these two models, we show that our proposed methods can work with both small and large networks. The implementations are based on a public code source Mu (2020).

## 4.2 RESULTS ANALYSIS

We first evaluate our methods on Mini-ImageNet using Convnet4 as the backbone model. We compare our method with previous state-of-the-art models including Meta-Learner LSTM Ravi & Larochelle (2016), TAML Jamal & Qi (2019), MAML Finn et al. (2017), Meta-SGD Li et al. (2017), MetaNet Munkhdalai & Yu (2017), Matching Network Vinyals et al. (2016), Relation Network Sung et al. (2018), Prototype Network Snell et al. (2017), and GNN Garcia & Bruna (2017). The experiments are conducted under two settings: 5-way-1-shot and 5-way-5-shot. The comparison results are summarized in Table 1.

From the results, we can observe a major performance improvement after applying our LWDGH. Comparing to the baseline model MAML, the performances are improved by 2.63% and 3.95% in

Table 2: Test accuracy for 5-way classification on Tiered-ImageNet. MAML's results are reproduced based on the published code.

| Task | Backbone | MAML | MAML+LWDGH |
|------|----------|------|------------|
| 5-way-1-shot | Convnet4 | 50.10±1.97 | 53.63±1.80 |
| 5-way-1-shot | Resnet12 | 51.97±1.76 | 60.93±1.91 |
| 5-way-5-shot | Convnet4 | 67.16±0.91 | 67.75±0.97 |

5-way-1-shot classification and 5-way-5-shot classification tasks, respectively. Notably, our method outperforms all optimization-based and metric-based models on the 5-way-1-shot classification task. This demonstrate that our method enables model to learn effectively even with limited training examples. On the 5-way-5-shot classification task, the proposed method outperform most of state-of-the-art models. Also, the proposed method outperforms three main gradient manipulation methods under the multi-task learning setting.

## 4.3 RESULTS USING A LARGE-SCALE DATASET

Previously, we evaluated our methods on the Mini-ImageNet dataset using Convnet4 as the backbone model. In this section, we conduct experiments using a large-scale dataset to further verify the effectiveness of our methods. To this end, we use the Tiered-ImageNet dataset, which contains 779,165 images. The experiments are conducted under three settings: 5-way-1-shot on Convnet4, 5-way-1-shot on Resnet 12, and 5-way-5-shot on Convent4. The results are summarized in 2. The results show that the model trained with our methods outperforms baseline models by 3.53%, 8.96%, and 0.69% on three settings, respectively. Notably, the performance improvements are much larger in fewer-shot experiment settings, which shows that our methods can effectively deal with noisy gradient updata and better train the meta-model.

## 4.4 RESULTS USING A LARGE BACKBONE MODEL

Previously, we evaluate our method using Convnet4, which is a relatively small network. To show the performances on large models, we change the backbone to Resnet12, which contains about 8 million trainable parameters. The results are summarized in Table 3. From the results, we can observe that our method outperforms MAML with a margin of 4.3%. Compared to results using Convnet4, the performance is further improved, which indicates that

Table 3: Comparison results for 5-way-1-shot classification on the Mini-ImageNet dataset using Resnet12.

| Method | Type | 5-way-1-shot |
|--------|------|--------------|
| DynamicNet | optimization | 58.55±0.50 |
| MAML | optimization | 54.87±1.82 |
| SNAIL | model | 55.71±0.99 |
| adaResnet | metric | 56.88±0.62 |
| MAML+LWDGH | optimization | **59.13**±2.01 |

our gradient harmonization method is more effective on backbone models. Compared to previous state-of-the-art models, our method outperforms them by a margin of at least 0.58%.

## 4.5 RESULT USING A SPECIAL-DESIGNED DATASET

Previous experimental studies are mainly based on Imagenet. The classes have significant differences from each other. It has been shown that our methods work well on such kinds of datasets with different backbone models. However, it is unknown if the proposed methods can perform well on

Table 4: Comparison results of Convnet4 based model on the Cub200 dataset using 5-way classification tasks.

| Method | 5-way-1-shot | 5-way-5-shot |
|--------|--------------|--------------|
| MAML | 54.00±1.78 | 61.99±0.95 |
| MAML+LWDGH | 58.13±1.73 | 68.11±0.97 |

datasets with similar features. This section conducts experiments to demonstrate the effectiveness of our methods on such kinds of datasets. To this end, we use the Cub200 dataset, which contains 11,788 images of 200 kinds of birds. The experimental settings are 5-way-1-shot and 5-way-5-shot classification tasks using Convnet4 as the backbone model. The results are summarized in Table 4. The result shows that the model trained with our methods outperforms MAML by 4.13% and 6.12% on 5-way-1-shot and 5-way-5-shot classification tasks, respectively, which shows that our methods enhance the learning efficiency and enable the model to extract advanced features to distinguish very similar objects.

## 4.6 ABLATION STUDY

We introduce a gradient harmonization method and an explore-exploit learning schedule to prevent the meta-model from over-emphasizing specific fine-tuned models. To evaluate the contributions of each component, we conduct ablation studies under the setting of 5-way-1-shot classifications on the Mini-ImageNet dataset. To

Table 5: Ablation study on each proposed component using the Mini-ImageNet dataset.

| Method | 5-way-1-shot | GCI |
|---|---|---|
| MAML | 48.17±1.75 | 4.8171 |
| MAML + LWDGH | 50.73±1.63 | 4.7545 |
| MAML + LWDGH + EE | 51.33±1.76 | 4.7146 |

this end, we run the experiments for 20 epochs and observe how the confliction changes. Each epoch contain 200 groups of tasks. And each group contains four 5-way-1-shot image-classification tasks leading to four gradients for each meta update. We train these task groups with MAML ,LWDGH and LWDGH+EE method separately and compute the GCI values.

The results are summarized in Table 5. We can observe that all components make significant contributions to the overall performance. In particular, the performance of LWDGH drops by 0.6% without using the explore-exploit scheduler, which indicates that the meta-model overemphasizes these fine-tuned models close to average gradient without sufficiently exploring the solution space. The proposed scheduler effectively balances explore and exploit, leading to a better-generalized meta-model. The results of GCI value are presented in the last two rows of Table 5. We can observe that with the original MAML setting. the average GCI score is 4.8171. After applying LWDGH, the confliction score drops to 4.7545. After applying EE, the confliction score can further drop to 4.7146. The result shows that our proposed method can effectively resolve the gradient confliction problem in meta-learning.

## 4.7 TRAINING EFFECTIVENESS STUDY

In Section 3.3, we introduce LWDGH aimed at dynamically resolving gradient conflicts arising from various fine-tuned models. It is anticipated that this approach will facilitate a more effective update of the meta-model, leading to quicker convergence. To illustrate this, we perform experiments and document the training losses for the models using both MAML and our LWDGH. The comparative results are depicted in Figure 4. This figure indicates that the model trained with LWDGH converges significantly faster than the one trained with MAML during the initial 50 epochs. This acceleration is particularly noticeable within the first 15 epochs, suggesting that the model benefits from the resolution of gradient conflicts. This property is also highly advantageous in meta-learning, where fast adaptation to new tasks is required.

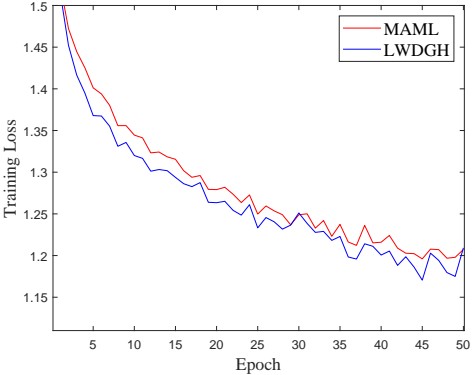

Figure 4: Training loss analysis. The red curve is for MAML and the blue curve is for our gradient harmonization.

## 4.8 HYPER-PARAMETER STUDY

One important hyper-parameter in our proposed explore-exploit scheduler is the transition epoch as described in Section 3.4. We conduct experiments to investigate how the change of transition epoch will affect the model performance. The experiments are conducted on the 5-way-1-shot classification on the Mini-ImageNet dataset. The backbone model is Convnet4. We evaluate two transition modes: fixed and cosine-annealing. In the fixed mode, we switch between exploration and exploitation at a predetermined number of epochs. Instead, cosine-annealing follows the mechanism described in Section 3.4 to

Table 6: Accuracy performance of proposed algorithm with different transition epoch.

| Transition Epoch | 5-way-1-shot |
|---|---|
| 20%-fixed | 50.43±1.96 |
| 20%-Cosine-Annealing | 49.67±1.75 |
| 33%-fixed | 50.80±1.68 |
| 33%-Cosine-Annealing | 51.33±1.76 |
| 50%-fixed | 50.53±1.82 |
| 50%-Cosine-Annealing | 50.67±1.77 |

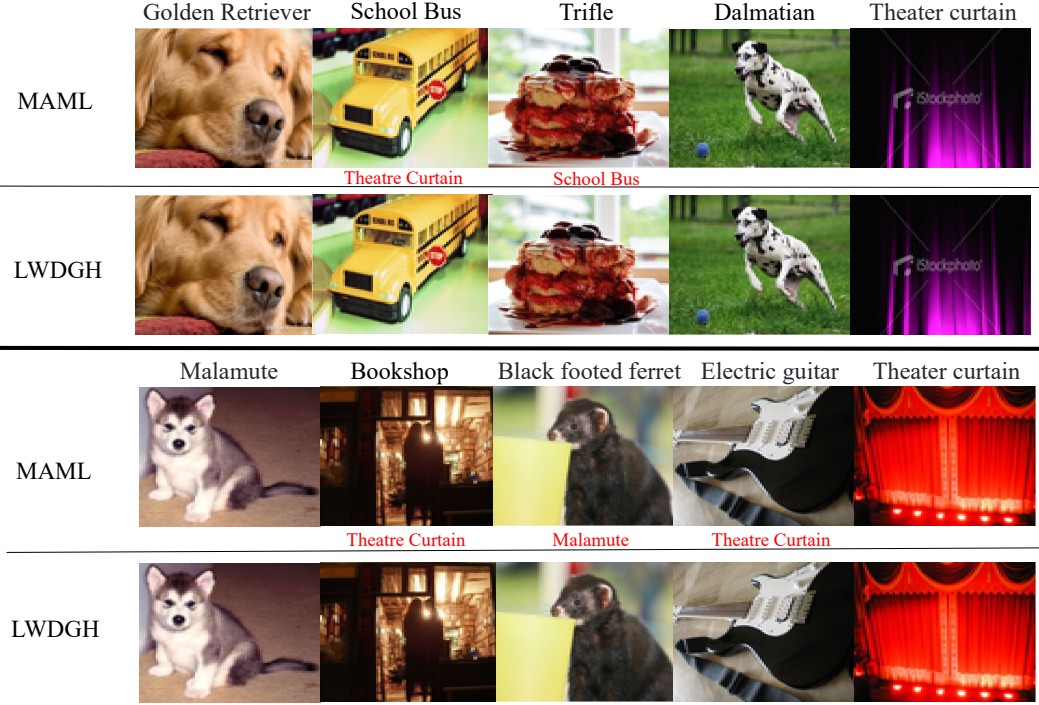

Figure 5: Examples of MAML predictions and our gradient harmonization predictions. The wrongly classified figures are colored red with predictions under them.

switch between explore and exploit. The selected transition epochs are 20%, 33%, and 50% of the total epoch. The results are summarized in Table 6. From the results, we can observe that when the transition epoch is 33% of the total epoch, the model performs the best. If the transition epoch is 20%, the transition happens too soon, preventing adequate exploration of the solution space. If the transition epoch is 50%, the transition happens too late, and the update conflicts have already impacted the model. And in both 33% and 50% scenarios, the results show that the cosine-annealing mechanism helps to improve the performance.

### 4.9 ERROR ANALYSIS

We visualize some predictions to show how our method can correct mistakes made by MAML. We sample two specific tasks from the test set and visualize the predictions in Figure 5. The model trained by MAML misclassifies 5 out of 10 samples across the two tasks. We observe that it extracts only limited knowledge for each class. For instance, a trifle is misclassified as a school bus since they share similar yellow and red colors. The black-footed ferret is misclassified as a Malamute due to their comparable body shapes. For tasks requiring advanced features, the MAML-trained model is insufficient. For example, the green background of the school bus image, the dark part of the bookshop image, and the shooting angle of the guitar image are quite misleading, causing the model to classify them into unrelated classes. By addressing the gradient conflict, our method enhances training efficiency, enabling the model to classify complex and misleading images.

## 5 CONCLUSION

In this paper, we address the gradient conflict issue in optimization-based meta-learning. To mitigate this issue, we propose a weighted gradients aggregation method that emphasizes models whose gradients align closely with the average gradient. Moreover, we introduce a layer-wise dynamic gradients aggregation technique using an attention operator, which dynamically assigns weights at each model level. Importantly, we develop an 'explore-exploit' mechanism to balance the weights assigned to different models, preventing any single fine-tuned model from being excessively emphasized. Experimental results demonstrate the efficacy of our approach in reducing task conflicts, outperforming both baseline and other state-of-the-art methods.

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
