# OpenReview forum: "Cross-Task Gradient Harmonization for Meta-Learning"
_ICLR.cc/2024/Conference — ICLR 2024 Conference Withdrawn Submission_

### Official Review · Reviewer_2jNY · 2023-10-17

**Soundness:** 1 poor
**Presentation:** 1 poor
**Contribution:** 1 poor
**Rating:** 3
**Confidence:** 5

**Summary:**

This paper tries to solve the gradient conflict problem in optimization-based meta-learning, such as MAML, by proposing a cross-task gradient harmonization method. Generally, from my perspective, the writing of paper is poor. Meanwhile, I think the method proposed in this paper is wrong. There are many problems in the formulations of this paper.

**Strengths:**

The empirical results are good.

**Weaknesses:**

__Main concerns:__
- __[Poor writing]__ The writing requires more polish work. Some sentences are hard to read and quite confusing.
- __[Unclear motivation]__ The motivation of the paper is not clear.
- __[Incorrect formulation]__ The formulation of problem is poor and incorrect. For example, although it is widely accepted that $\boldsymbol{\theta}$ denotes model paramaeters, it is necessary to define it before the method. However, it is not defined in Eq. (1). Besides, from my perspective, Eq. (1) means the update of model paramater $\boldsymbol{\theta}$, instead of the gradients.
- __[Wrong method]__ The core calculation seems wrong. (cf. Eq. (3)).

__Minor concerns:__
- The format reference do not follow the instructions of ICLR. The command `\citep` or `\citet` should be used. In this paper, many references use `\citet` or `\cite`, which drops the parentheses.
- I think that a set of legends are required in Fig. 1 to illstrate the meaning of each notation.
- From my perspective, Fig. 1 is not an 'example' of gradient conflict, it is an 'illustration' of gradient conflict.
- To be more clear, it would be better to write the concrete expression of $\boldsymbol{\bar{g}}$.

**Questions:**

1. In fact, until I finish reading section 3.1, I do not quite understand the definition of "gradient conflict in optimization-based meta-learning". So, could you please provide an definite definition of the problem you studied in this paper?
2. In my opinion, the gradient conflict in multi-task learning derives from the differences among tasks. However, in meta-learning, tasks are with the same format, such as N-way K-shot image classification tasks. So, why do you think that gradients from different tasks result in conflict? Could you provide some intuitive or theoretical evidence to prove the conflict?
3. Why $\sum_{T_i}||\boldsymbol{g}_i||(1-cos(\boldsymbol{g}_i, \boldsymbol{\bar{g}}))=\frac{||\boldsymbol{g}_i||-\boldsymbol{g}_i\cdot\boldsymbol{\bar{g}}}{||\boldsymbol{\bar{g}}||}$? It would be better to check the calculation.
4. What does Fig. 2 mean? What does the red line and blue points mean?
5. What are the relationships between $\boldsymbol{g}_q$, $\boldsymbol{g}^\ell$ and $\boldsymbol{g}^{\ell}_{q}$?
6. I do not think Eq. (4) is an attention operation, it is just a softmax operation. Note that in typical attention, given $q\in \mathbb{R}^{N\times D}$, $k\in \mathbb{R}^{M\times D}$ and $v\in \mathbb{R}^{M\times D}$, the operation is performed as ${\rm softmax}(\frac{q\cdot k^{\top}}{\sqrt{D}})\cdot v$. You'd better rethink about your formulation.

__Minor questions:__
1. Why $g_i$ $\bar{g}$ is used in ${\rm GCI}$ but $\boldsymbol{g}_i$, $\boldsymbol{\bar{g}}$ is used in the right part of Eq. (3)?
2. The right side of Eq. (3) lacks $\sum_{T_i}$?

---

### Official Review · Reviewer_xtbH · 2023-10-29

**Soundness:** 1 poor
**Presentation:** 2 fair
**Contribution:** 2 fair
**Rating:** 3
**Confidence:** 5

**Summary:**

The paper focuses on the problem of gradient conflict in the context of gradient-based meta-learning (GBML). A new method is introduced, named dynamic gradient harmonization. The method consists of measuring the gradient conflicts using a newly proposed metric (gradient conflict index) and then reweighting the gradients from tasks (and layers) accordingly. The method additionally includes an explore-exploit mechanism to obtain stronger performance.

**Strengths:**

* A new intuitive metric to measure the problem is introduced.
* The method leads to quite large improvements for MAML.
* The method is evaluated across a variety of classification datasets, including larger ones.
* Ablation study is provided to analyse the impact of explore-exploit mechanism, as well as a study of an important hyperparameter: transition epoch.
* The method appears novel.

**Weaknesses:**

* The idea of Gradient harmonization is likely to generalize to any gradient-based meta-learning (GBML) method, but it is only evaluated on the most basic one: MAML. This is a serious limitation given that there are many GBML methods.
* The evaluation is a bit inconsistent: e.g. 5-way 5-shot with ResNet12 is not evaluated in Table 2, or also 5-way 5-shot ResNet12 on MiniImageNet. Is there a particular reason to exclude these? ResNet models are commonly tested on the 5-way 5-shot scenario on those datasets/architecture.
* There are quite many typos/simple grammar mistakes in the paper and generally the writing could be more fluent. Example typos: K instead k in Section 2.2, “this work propose to”, “these method will”, “weighted gradient aggregation method, which aggregate gradients”, “The performances are improved by 2.63”, while the actual improvement is 3.23.
* It would be good to have the full evaluation for ablation - not only 20 epochs as it is not clear why that precise number.
* It would be valuable to evaluate the method also on cross-domain few-shot learning (e.g. on Meta-Album [A]), and given that gradient conflicts were originally studied in multi-task learning, multi-task few-shot learning would be particularly relevant (e.g. Meta Omnium [B]).
* Given that few-shot learning is still an active area of research, one would expect to see also more recent references in the related work as well as in the experimental evaluation.
* Many of the more advanced or recent approaches are left out from the experimental evaluation, so that the results can give an impression the result achieved is state-of-the-art. For example, Meta-Curvature is a relatively popular method from 2019 that outperforms the proposed approach and there are many more. I would recommend that the authors show Gradient harmonization can be added to a variety of GBML methods and then comparison with the most recent methods would not be needed

[A] Meta-Album: Multi-domain Meta-Dataset for Few-Shot Image Classification (NeurIPS’22)

[B] Meta Omnium: A Benchmark for General-Purpose Learning-to-Learn (CVPR’23)

**Questions:**

* Does Gradient harmonization help improve performance of other GBML methods in addition to MAML? (would be good to show it on 3-5 common GBML methods)
* How well does the method work in cross-domain and multi-task few-shot settings?
* How are the performances in the missing scenarios?

---

### Official Review · Reviewer_1XqW · 2023-10-31

**Soundness:** 2 fair
**Presentation:** 2 fair
**Contribution:** 2 fair
**Rating:** 3
**Confidence:** 5

**Summary:**

This paper proposes a regularization technique for optimization-based meta-learning frameworks by measuring the gradient conflicts between tasks. To this end, the author proposes a measurement called, GCI, which measures the cosine distance between the averaged gradient and each task multiplied by the gradient norm. They further consider layer-wise gradient rescaling and suggest using a learning rate scheduler. They verify the effectiveness under few-shot learning scenarios by using mini-imagenet, tiered-imagenet, and CUB datasets.

**Strengths:**

**Strengths**

(1) The overall writing is clear. Improving the experiment section writing will further improve the overall presentation.

(2) The proposed method is sound.

**Weaknesses:**

**Weaknesses**

(1) Improving the base MAML may not be interesting to the community, as by using some engineering techniques, MAML outperforms the current performance in the paper (See Table 2 in MAML ++ [1]). Furthermore, MAML++ has already visited the use of a learning rate scheduler.

(2) Although the gradient alignment (or confliction) that is proposed in [2] is different (as they sparsify the gradients based on support and query set of the same task), it is worth discussing and comparing.

(3) The overall performance is quite low compared to recent regularization for meta-learning [3,4]. Furthermore, the baselines are quite outdated, i.e., there are more recent works that show much better results in mini-ImageNet [5,6,7].

(4) Also, the author only considered MAML while there exist several optimization-based meta-learning schemes [8,9]. I think it is worth showing that the proposed method consistently improves these methods as they are also optimization-based meta-learning.

(5) Improving optimization-based meta-learning for few-shot learning may not be an interesting direction, since, other methods (e.g., metric-based, amortization-based, pretraining, or self-supervised learning) outperform optimization-based meta-learning in large-scale benchmarks as well [10,11,12]. Maybe considering some applications where optimization-based meta-learning is truly important will be better.

(6) The novelty is somewhat limited. The importance of using gradient conflict when handling multiple tasks (e.g., multi-task learning) is quite well-known in the field [13]. Also, using layer-wise gradient computation and learning rate scheduling is also known for improvement [1].

(7) The experiment section organization can be improved.
- Missing experiments of 5way 5shot in ResNet12 (note that most papers show this case as well).
- showing cross-domain adaptation will be interesting, e.g., meta-train on mini-ImageNet and meta-test on CUB200.
- Ablation study of using layer-wise GCI will be interesting.

**Reference**\
[1] How to train your MAML, ICLR 2019\
[2] Gradient sparsity in meta and continual learning, NeurIPS 2021\
[3] Towards Enabling Meta-Learning from Target Models, NeurIPS 2021\
[4] Meta-Learning with Self-Improving Momentum Target, NeurIPS 2022\
[5] How to Train Your MAML to Excel in Few-Shot Classification, ICLR 2022\
[6] Meta-learning with differentiable convex optimization, CVPR 2019\
[7] Meta-Learning with Latent Embedding Optimization, ICLR 2019\
[8] Continuous-Time Meta-Learning with Forward Mode Differentiation, ICLR 2022\
[9] Bootstrapped Meta-Learning, ICLR 2022\
[10] Pushing the Limits of Simple Pipelines for Few-Shot Learning, CVPR 2022\
[11] Fast and Flexible Multi-Task Classification Using Conditional Neural Adaptive Processes, NeurIPS 2019\
[12] Learning a Universal Template for Few-shot Dataset Generalization, ICML 2021\
[13] Recon: Reducing conflicting gradients from the root for multi-task learning, ICLR 2023

**Questions:**

Can the author report the Pearson correlation of Figure 2? Increasing the number of tasks will have a more robust evaluation when measuring the correlation coefficient.

While I think showing the correlation is good for the motivation, it does not tell that the suggested metric is the optimal way of measurement.

---

### Official Review · Reviewer_AKfw · 2023-11-01

**Soundness:** 2 fair
**Presentation:** 3 good
**Contribution:** 2 fair
**Rating:** 3
**Confidence:** 4

**Summary:**

This paper introduces a new metric, the Gradient Conflict Index (GCI), which weights different meta-gradients to harmonize potentially conflicting gradients. Various experiments demonstrate the effectiveness of the proposed algorithm.

**Strengths:**

- The paper is well-structured.
- The paper proposes a novel metric for quantifying gradient conflict that arises between multiple tasks.
- The paper conducts a variety of experiments to showcase the effectiveness of the proposed algorithm.

**Weaknesses:**

- The authors highlight the disparity of the gradient conflict issue between the context of multi-task learning and meta-learning. However, they propose the method based on the observation from multi-task learning without justification in the context of meta-learning.
	- For example, while the performance damage of gradient conflicts has been well-studied in multi-task learning, its effect on meta-learning remains uncertain. However, this paper fails to demonstrate whether such conflicts are a significant issue in meta-learning. Given that meta-learning assumes tasks conform to the same meta distribution and meta-learning algorithms learn shared knowledge across tasks, gradient conflicts do not appear to be a critical issue.
	- Another example is that in Sec 3.3, the authors introduce layer-wise attention based on observations from SHI et al. (2023) without justification in the context of meta-learning.


- The calculation of GCI involves the gradient magnitudes. Gradient magnitude has a strong correlation with the loss.  Consequently, the relation between loss and GCI is affected by both the magnitude of the gradient and the diversity of gradients. Further clarification is required to elucidate the precise relationship between gradient diversity and the loss or performance.

- The gradient harmonization method prioritizes or up-weights task gradients that closely align with the original average update gradient, resulting in a new update gradient that is expected to have a similar direction as the original update gradient but with a different magnitude. From this perspective, the proposed method adjusts the update gradient magnitude, which has an effect similar to tuning the learning rate while using the original update gradient. Interestingly, the authors use an explore-exploit learning schedule inspired by a learning rate schedule.

- The proposed method is a task reweighting framework, but it is not discussed and compared with existing works on task reweighting, such as [1-3]
[1] Li, Xiaomeng, et al. "Difficulty-aware meta-learning for rare disease diagnosis." Medical Image Computing and Computer Assisted Intervention–MICCAI 2020. Springer International Publishing, 2020.
[2] Sun, Qianru, et al. "Meta-transfer learning for few-shot learning." Proceedings of the IEEE/CVF conference on computer vision and pattern recognition. 2019.
[3] Yao, Huaxiu, et al. "Meta-learning with an adaptive task scheduler." Advances in Neural Information Processing Systems 34 (2021): 7497-7509.


- It fails to demonstrate the effectiveness of the gradient harmonization by
	- mathematically showing how $g_u^\ell$ reduces GCI
	- empirically showing the performance with only gradient harmonization and no learning schedule
	- showing how $g_u^\ell$ improves the model update


- About Writing
	- Sec 4.9 only proves that LWDGH outperforms MAML. it does not directly prove how the proposed method improves performance.
	- Step 12: The gradient equation is confusing. It would be better to use a meta-gradient equation similar to Eq (2).
	- ResNet12 is not 18-layer.

**Questions:**

Please address the issues in the weaknesses.

---

### Official Review · Reviewer_cm6w · 2023-11-01

**Soundness:** 2 fair
**Presentation:** 2 fair
**Contribution:** 2 fair
**Rating:** 3
**Confidence:** 5

**Summary:**

This paper proposes Dynamic Gradient Harmonization to address the gradient conflict issue in optimization-based meta-learning. Specifically, the authors propose to weight the aggregation of gradients from fine-tuned models with an attention mechanism to weight the gradients from different tasks.  The authors also further propose an explore-exploit mechanism to further improve the model performance. Experiments on several datasets and architectures show the effectiveness of the proposed methods.

**Strengths:**

* This paper proposes Dynamic Gradient Harmonization to address the gradient conflict issue in meta-learning.

* The paper is easy to follow.

**Weaknesses:**

* In the related work section, the authors argue that “Under meta-learning setting, these method will be less effective since they are designed to optimize a fixed set of tasks not a general objective.” However, the reviewer does not agree with this view point, since the fine-tuned model also has a gradient, then views each task in a meta-batch, the methods in existing multi-task learning can still be applied to the meta-training process.


* There is a lack explanation of the proposed explore-exploit learning scheduler. Why was the scheduler designed in such a way?  What are the benefits it brings to meta-learning?



* Existing works on meta-learning with multi-objective optimization have already dealt with gradient conflicts in meta-learning [1].  This method should be discussed in related works and compared in experiments.  Further, this also decreases the novelty of this work since they address the similar problem with similar approaches.


Reference:

Multi-Objective Meta Learning. NeurIPS 2021

**Questions:**

N/A

---

### Official Review · Reviewer_zD2n · 2023-11-03

**Soundness:** 3 good
**Presentation:** 3 good
**Contribution:** 2 fair
**Rating:** 5
**Confidence:** 4

**Summary:**

This method improves optimization-based meta-learning methods by addressing gradient conflicts.  It weights the gradient for each task according to how much it matches the average batch gradient.  This resolves much of the disagreement by choosing a direction that optimizes at least some of the sampled tasks (after the step, another random batch of tasks later on may move in a direction optimizing a down-weighted task).  In addition to the harmonization method, this paper describes a metric for measuring grad conflict ("GCI"), as well as an explore-exploit-style learning schedule that makes use of both original average gradients (explore) and harmonized gradients (exploit).  Measurements are performed on miniImagenet, tieredImagenet and CUB, finding improved performance compared to vanilla MAML as well as other relevant meta-learning baselines (in the case of miniImagenet).

**Strengths:**

This is a simple method with what appears to be a nice improvement.  It's described clearly, with experiments demonstrating gains in multiple comparison settings.

**Weaknesses:**

The harmonization technique is very similar to Eshratifar et. al. NIPS meta-learning workshop 2018 (arxiv:1810.08178).  It is different in that the weight values are exponentiated here, where as in that work they aren't, and that agreements are measured layerwise.  It would be good to compare to this as well.

The ablations of this method are also not yet very deep.  What is the effect of applying layerwise vs to all layers?  What about the softmax temperature for the weights?  While I appreciate the different configurations already examined and the study on EE mixing, I think these other components would also be good to explore.

**Questions:**

Fig 2:  I don't see how a relationship between loss and GCI necessarily corresponds to reduced gradient conflicts.  Since the magnitude $||g_i||$ is in the definition, larger $||g_i||$ results in larger values for GCI --- in fact, this is mentioned in the text.  Since $g_i = \nabla L$, larger $L$ results in linearly larger $||g_i||$, which is also what is seen in the figure.

Sec 4.6:  Since the rows are titled "LWDGH" and "LWDGH + EE" in this section, it's unclear to me whether EE was applied in the experiments before in secs 4.1 .. 4.5.  Was it?

Sec 4.8 / tabel 6:  This might also compare to a uniform random sampling strategy at the same ratios, i.e. $explore = rand() < p$.

Eq 3:  sum over T_i is in the middle part of the equation, but not sure if it should be, it doesn't seem to be in the other expressions.